# Ultrasonic Beamforming-Based Visual Localisation of Minor and Multiple Gas Leaks Using a Microelectromechanical System (MEMS) Microphone Array

**DOI:** 10.3390/s25103190

**Published:** 2025-05-19

**Authors:** Tao Wang, Jiawen Ji, Jianglong Lan, Bo Wang

**Affiliations:** School of Automation, Beijing Institute of Technology, No. 5 South Zhongguancun Street, Haidian District, Beijing 100081, China

**Keywords:** gas leak localisation, imaging, ultrasonic sensor array, beamforming

## Abstract

**Highlights:**

**What are the main findings?**
A universal, non-contact ultrasonic imaging method was developed for real-time gas leak detection and localisation in pressure vessels, pipelines, valves, and connectors.The proposed system achieved high-precision localisation (0.68 cm error at 1 m) and detected minor leaks as small as 24 mL/min while distinguishing multiple leak sources.

**What is the implication of the main finding?**
The method enhances industrial and environmental safety by providing an effective, scalable solution for detecting and visualising gas leaks.Its integration into embedded systems enables real-time monitoring, supporting predictive maintenance and reducing the risk of hazardous incidents

**Abstract:**

The development of a universal method for real-time gas leak localisation imaging is crucial for preventing substantial financial losses and hazardous incidents. To achieve this objective, this study integrates array signal processing and electronic techniques to construct an ultrasonic sensor array for gas leak detection and localisation. A digital microelectromechanical system microphone array is used to capture spatial ultrasonic information. By processing the array signals using beamforming algorithms, an acoustic spatial power spectrum is obtained, which facilitates the estimation of the locations of potential gas leak sources. In the pre-processing of beamforming, the Hilbert transform is employed instead of the fast Fourier transform to save computational resources. Subsequently, the spatial power spectrum is fused with visible-light images to generate acoustic localisation images, which enables the visualisation of gas leak sources. Experimental validation demonstrates that the system detects minor and multiple gas leaks in real time, meeting the sensitivity and accuracy requirements of embedded industrial applications. These findings contribute to the development of practical, cost-effective, and scalable gas leak detection systems for industrial and environmental safety applications.

## 1. Introduction

The scale of gas storage and transportation, including natural gas transportation and carbon capture and storage, has expanded progressively with industrial development, leading to the increased use of pressure vessels and pipelines. However, these containers are easily damaged by corrosion, abrasion, and third-party damage [1,2], leading to gas leaks that cause significant financial losses and hazardous incidents. Gas pipeline leakage has been extensively researched, often relying on sensors directly mounted on the pipeline to detect the propagation of gas leak signals on the pipe wall for leak localisation. However, gas leaks can also occur in pressure vessels [3]. Bolted flange joints and valves are particularly susceptible to leaking [4]. Therefore, the development of a universal, non-contact method for detecting and locating gas leaks is crucial.

Leakage sources generate a broad range of frequencies; however, 40 kHz has been identified as the optimal frequency for maximising the signal-to-noise ratio. This selection minimises ambient noise interference while ensuring the reception of the strongest ultrasonic leak signals [5]. Considering accuracy, manufacturing costs, operational complexity, and device simplicity, ultrasonic detection techniques [6] are the most suitable for large-scale applications. They offer advantages over other gas leak detection methods, such as differential pressure methods [7], infrared imaging methods [8], and mass spectrometry methods in terms of efficiency and practicality [9].

However, numerous challenges persist in their practical applications. One key issue is the applicability of ultrasonic detection methods for detecting multiple gas leak sources. As secondary leaks are highly likely during real-world detection, the system must be capable of accurately detecting and localising multiple leak sources simultaneously. Traditional ultrasonic detection methods often establish sound models of gas leaks in the time domain [3,5]. The time difference of arrival (TDOA) algorithm is a representative example within the time-domain methods. Mengjie et al. [10] combined envelope detection and traditional cross-correlation algorithms to determine the time differences between gas leak sources arriving at different sensors in a sensor array, thereby enabling the localisation of leaks using the TDOA algorithm. Subsequently, the TDOA algorithm was integrated with the energy decay (ED) algorithm to further improve positioning accuracy [11]. However, these methods showed limited performance in scenarios involving multiple gas leaks. Owing to the similarity of the leak sources, the possibility that the ultrasonic signal characteristics of multiple gas leaks overlap in the time domain and become unextractable is high, which poses obstacles to establishing a comprehensive model of multiple gas leak sources in the time domain.

Second, the applicability of ultrasonic detection methods for detecting minor gas leaks is problematic. Although establishing a multiple gas leak sound model in the time domain is challenging, a spatial-domain acoustic field model can address the issue of multisource localisation [12,13]. However, source localisation algorithms often focus on low-frequency, high-intensity audible sound sources [14,15]. The acoustic signals of minor gas leaks are extremely weak and can be easily overwhelmed by ambient noise. Therefore, leak acoustic signals must be specifically pre-processed, such as selecting a frequency of 40 kHz for gas leak detection [16]. Not only is the frequency band different, but the intensity of the acoustic signals of a minor gas leak is also significantly lower. Thus, applying source-localisation algorithms to ultrasonic detection imposes higher requirements on the processing of leaked acoustic signals.

Finally, achieving real-time performance of ultrasonic detection methods is challenging. In the development of localisation algorithms, researchers often place more focus on direction-of-arrival (DOA) estimation accuracy than on real-time performance. Although high-resolution DOA estimation algorithms significantly enhance sound source discrimination capabilities, even their computationally optimized versions remain significantly longer, processing latency compared to conventional beamforming [17], making them better suited for static laboratory experiments rather than portable field detection. Achieving a balance between estimation accuracy and computational efficiency requires algorithmic complexity reduction, which bridges the gap between theoretical models and real-world detection.

To address these challenges, this study proposed an ultrasonic beamforming-based imaging method for gas leak localisation on embedded hardware. Section 2 describes the processing of gas leak acoustic signals to resolve the issue of minor and multiple gas leak localisations and visualisations. Section 3 aims to design an array with superior noise suppression capabilities to enhance the system’s sensitivity to minor gas leaks. Section 4 presents an embedded solution for a portable device and shows leak detection and localisation experiments on this system. A series of experiments demonstrated that the proposed method achieved remarkable detection and a high-precision localisation of minor single-point gas leaks, stable multi-angle detection and localisation of multiple gas leaks, and distinction between closely spaced dual gas leak sources. Additionally, it demonstrates a certain level of real-time performance in an embedded system.

## 2. Materials and Methods

### 2.1. Localisation Algorithm

Owing to its simplicity, speed, robustness, and superior performance at high frequencies, the conventional beamforming (CBF) method is highly suitable for gas leak detection and localisation in the ultrasonic domain.

First, a multisource gas leak signal model was established based on the far-field plane-wave assumption. When there are *K* gas leak sound sources, the signal collected by the *m*th element of the array can be expressed as(1)$$y_{m}\left(t\right)=\sum_{k=1}^{k=K}S_{k}\left(t\right)e^{-j2\pi (x_{m}\;sin\theta_{k}\;cos\phi_{k}+y_{m}\;sin\theta_{k}\;sin\phi_{k})/\lambda }\;(m=1,2,\ldots ,M)$$
where $S_{k}\left(t\right)$ is the ultrasonic signal from the *k*th gas leak source arriving at the array centre, ($x_{m},\;y_{m})$ are the coordinates of the *m*th element of the sensor array, $\left(\theta_{k},\phi_{k}\right)$ are the specific incidence angles of the *k*th gas leak source; and λ represents the wavelength of the ultrasonic waves emitted by the gas leak source. The number of array elements is described as M. From Equation (1), the signal collected by the *m*th element of the array is expressed as a linear superposition of the spatial information from each gas leak sound source.

In the beamforming algorithm, the spatial information of the gas leak sources is processed through angular scanning. The scanning angle $\left(\theta ,\phi \right)$ is defined as shown in Figure 1.

From Figure 1, the differences in path lengths Δ*d* and phase differences Δφ are formulated as follows:(2)$$\Delta d=\left(x_{m}cos\emptyset +y_{m}sin\emptyset \right)sin\theta$$(3)$$\Delta \varphi =\frac{2\pi \Delta d}{\lambda }=\frac{2\pi }{\lambda }(x_{m}sin\theta cos\emptyset +y_{m}sin\theta sin\emptyset )$$

Based on the phase differences, the steering vector is defined as:(4)$$a\left(\theta ,\;\phi \right)=\left[\begin{array}{c} \begin{array}{c} e^{j\frac{2\pi }{\lambda }(x_{1}sin\theta cos\phi +y_{1}sin\theta sin\phi )}\\ e^{j\frac{2\pi }{\lambda }(x_{2}sin\theta cos\phi +y_{2}sin\theta sin\phi )}\\ \ldots \ldots \end{array}\\ e^{j\frac{2\pi }{\lambda }(x_{M}sin\theta cos\phi +y_{M}sin\theta sin\phi )} \end{array}\right]\;(\theta ,\phi )\in \Theta$$
Here, Θ denotes the set of all scanning angles, commonly known as the field of view. Equation (4) shows that the steering vector is determined by the scanning angle and microphone coordinates in the array. However, this is not affected by other factors. The spatial information of the signals is processed by multiplying the steering vector by the array signals collected. The beamformer output is expressed as(5)$$z\left(t,\theta ,\;\phi \right)=a^{H}\left(\theta ,\;\phi \right)Y\left(t\right)=a^{H}\left(\theta ,\;\phi \right)\left[\begin{array}{c} y_{1}\left(t\right)\\ \begin{array}{c} y_{2}\left(t\right)\\ \ldots \\ y_{M}\left(t\right) \end{array} \end{array}\right]\;(\theta ,\phi )\in \Theta$$
where $Y\left(t\right)$ is a matrix composed of signal samples collected from all array elements. When the delay factors within the steering vector for a specific scanning angle $\left(\theta ,\phi \right)$ accurately align with the phase delays in the spatial information of a particular gas leak source $\left(\theta_{k},\phi_{k}\right)$, the beam power is enhanced. The beam power is calculated as:(6)$$P\left(\theta ,\;\phi \right)=E\left\{{z\left(t,\theta ,\;\phi \right)}^{2}\right\}=a^{H}\left(\theta ,\;\phi \right)\cdot R\cdot a\left(\theta ,\;\phi \right)$$
where E is the statistical expectation, and R is the covariance matrix of the array signal. This matrix captures the spatial characteristics and statistical dependencies of the signals collected by the array elements and is defined as:(7)$$R=E\left\{Y(t)Y^{H}(t)\right\}$$

The beam power calculated using Equation (6) is a two-dimensional matrix that depends on the scanning angle. Physically, it represents the spatial spectral power density of gas leak sources. Due to the linear superposition of spatial information from multiple gas leaks, as defined in the signal model, the distribution of multiple leak sources can be obtained after beamforming.

### 2.2. Signal Processing Method

Microelectromechanical system (MEMS) microphones, which include a substantial number of integrated circuits for signal conditioning and other functions within the same sensor package, are widely used in acoustic applications [19]. MEMS microphones are classified as analogue or digital, depending on their output type. To reduce the complexity of the external signal conditioning circuitry, digital MEMS microphones have become a preferred choice in the design phase of large arrays. Moreover, digital MEMS microphones are immune to radio frequency (RF) noise and exhibit reduced sensitivity to electromagnetic interference compared to analogue MEMS microphones. The encoded output format of digital MEMS microphones is pulse-density modulation (PDM), which represents an oversampled 1-bit audio signal.

In the digital signal processing system, the PDM data format generated at a high sampling rate must be converted to the pulse code modulation (PCM) format. In theory, direct low-pass filtering or decimation is theoretically sufficient. However, such approaches would increase the complexity of the filter design, thereby increasing the computational burden of decoding. Consequently, cascaded integrator-comb (CIC) filters are commonly employed. CIC filters comprise integrators, decimators, and comb filters; multipliers are not used, which results in minimal resource consumption, high computational speed, and high-speed filtering. Given the linearity of the filters, the order of the components can be rearranged. Placing decimators prior to the comb filters reduces the required buffer depth of the delay buffers in the comb filters (Figure 2), thus further saving hardware resources.

In this study, the CIC filter used for decoding and reconstructing PDM signals was configured with a 5-stage cascade to improve filtering performance and signal reconstruction; a decimation factor of 24 was used to balance the filter complexity and performance, and a differential delay of 1 was the standard choice for optimising the comb filter response. The PDM sampling rate was 4.8 MHz, and the signal sampling rate after decimation was 200 kHz, suitable for capturing ultrasonic signals at approximately 40 kHz in accordance with the Nyquist–Shannon sampling theorem.

In industrial environments, machinery such as motors, compressors, and exhaust fans generate noise with energy predominantly concentrated below 20 kHz, whereas electronic circuit noise typically exhibits higher-frequency characteristics. Given the superior signal-to-noise ratio at 40 kHz [5], a 100th-order Chebyshev bandpass filter is selected for its exceptional bandpass isolation and out-of-band attenuation capabilities, which are critical for suppressing low-frequency mechanical noise and high-frequency circuit interference simultaneously. This filter order was specifically chosen to ensure sufficient stop-band attenuation, with its spectral response characteristics depicted in Figure 3.

To achieve precise phase delay, this study defined the steering vector in the frequency domain. Since sensor signals contain only real components, their frequency spectra exhibit symmetric positive and negative frequency components, which causes central symmetry in conventional beamforming (Figure 4a). Traditional methods employ a fast Fourier transform (FFT) to transform signals to the frequency domain for phase shifting, followed by weighted summation across frequency bands to obtain a spatial power spectrum [14,20]. However, this approach requires signal frame segmentation and imposes high resource consumption during multi-channel hardware synchronization.

In contrast, this study employed the Hilbert transform to introduce a 90° phase delay for real signals, directly constructing analytic signals that satisfy Equation (1) to suppress negative frequency components:(8)$$y(t)=u\left(t\right)+j*u(t)e^{-j\frac{\pi }{2}}$$
Here, $u\left(t\right)$ is the input signal and y(t) is the analytic signal constructed by Hilbert transform. Beamforming is ultimately realized through time-domain complex calculations, eliminating the central symmetry (Figure 4b). The Hilbert transform can be implemented by hardware-based FIR filters, supporting continuous input signals without frame segmentation. This approach minimizes resource consumption, making it conducive to the deployment of real-time applications.

As the order increases, the gain of the modulated signal fluctuates in both directions. Given that the gas leak signal is inherently weak, even a slight attenuation of approximately 40 kHz was unacceptable. Therefore, this study employed only a 10th-order FIR filter to implement the Hilbert transform, which maintains a positive gain near 40 kHz and requires minimal resources.

When gas is released through a leak in a container, the airflow near the leak generates turbulence. Turbulence is not a completely random, disordered motion but an organised motion within apparent disorder known as a quasi-ordered structure [21]. Hence, turbulence can be considered an intermittent process that alternates between periods of activity and quiescence (Figure 5). This characteristic reflects the development and breakdown of turbulent flow, resulting in a regularly shaped envelope [22]. Therefore, to capture the “active period” and “quiet period” information within the gas leak acoustic signals, the sample count should be maximised to reduce the estimation error of the covariance matrix R in Equation (7).

### 2.3. Imaging Method

The PDM output from the digital MEMS microphones on the array was converted to PCM by a CIC filter, and noise was filtered out using a Chebyshev bandpass filter. Subsequently, the signals were transformed into analytic signals via the Hilbert transform to obtain a spatial power spectrum, which characterised the distribution of multiple gas leaks through beamforming. Upon completing the beamforming calculations, the spatial information must be transformed into a visual representation, as described below.

To convert the spatial power spectrum into an image suitable for fusion with visible-light data, it first needs to be standardised as(9)$$S\left(\theta ,\;\phi \right)=10\log\left(\frac{P\left(\theta ,\;\phi \right)}{max\left(P\left(\theta ,\;\phi \right)\right)}\right)+DR$$
The dynamic display area of $S\left(\theta ,\;\phi \right)$ can be adjusted by increasing or decreasing the value of the dynamic range (DR).

Second, the dynamic display area needs to be transformed from spherical coordinates to Cartesian coordinates, which are associated with visible-light images. The r represents the distance between the device and the gas leak source.(10)$$\left\{\begin{array}{c} x=rsin\theta cos\phi \\ y=rsin\theta sin\phi \\ S\left(\theta ,\;\phi \right)⟹S(x,y) \end{array}\right.$$

Finally, the dynamic display area is mapped onto an image. It is first mapped onto an 8-bit depth greyscale image. The greyscale image is then converted into a colour image. Finally, an alpha channel for transparency is added using a mask, resulting in a semi-transparent colour image of multiple gas leaks. The overall processing pipeline from signal acquisition, pre-processing, beamforming, image fusion, to final visualization is shown in Figure 6.

This section presents an optimized signal processing approach for gas leak detection and localisation. By employing a cascaded filter bank, the approach decoded and reconstructed multi-channel digital microphone signals, performed noise suppression, and constructed analytic signals, thereby achieving frequency-domain beamforming from time-domain signals. The method achieved significant computational and resource efficiency, making it particularly suitable for implementation on portable embedded hardware. When integrated with image processing techniques, it enables the visualization of gas leak.

## 3. Array Design

The sensor array comprised 32 digital MEMS microphones (SPH0641LU4H-1, provided by Knowles). SPH0641LU4H-1 is a miniature, high-performance, low-power bottom-port digital silicon microphone with a single-bit PDM output (Figure 7a). Using Knowles’ proven high-performance MEMS technology, the device comprised an acoustic sensor, a low-noise input buffer, and a sigma–delta modulator, rendering it suitable for applications that demand excellent wideband audio performance and RF immunity. Moreover, it possesses omnidirectional pickup capability. It supports an ultrasonic mode, exhibiting a flat amplitude–frequency response curve centred around 40 kHz, as shown in Figure 7b, which effectively captured the ultrasonic signals of gas leaks. Thus, SPH0641LU4H-1 was selected as the preferred sensor for this system.

Array geometry, which refers to the position of microphones in an array, has been extensively studied. Digital MEMS microphones are compact, and their peripheral circuit design is straightforward, enabling greater design flexibility in an array design. To engineer the optimal array, the initial step involves quantifying the performance metrics of the array; these are the main beam width (BW), which signifies the spatial resolution of the beam, and the maximum dynamic range (DR), which represents the noise suppression capability of the array (Figure 8a). Currently, commonly used two-dimensional array structures include [23] the ring, Archimedean spiral, Dougherty log-spiral, Arcondoulis spiral, rose spiral, sunflower, underbrink array, and annular array [24]. Based on their design principles, the corresponding optimal array configurations were designed within a fixed aperture, and their performance metrics were evaluated.

Owing to the highly irregular distribution of the dynamic range across the frequency range of interest (30–50 kHz), establishing a single, dynamic range performance parameter is challenging. The average dynamic range is not a sufficient measure because a high dynamic range at some frequencies cannot compensate for the very low dynamic range values at other frequencies. Therefore, the variance of the dynamic range was proposed to determine whether the dynamic range is uniformly distributed across the spectrum. The results of the performance metrics obtained through simulations of different array configurations are listed in Table 1.

From Table 1, evidently, the subtle difference in the BW performance of these arrays in the high-frequency domain is imperceptible to the human eye. The annular array exhibited the largest dynamic range, demonstrating superior noise suppression and stability across the frequency range of interest. It is capable of flexibly and uniformly arranging microphones within the specified array aperture, overcoming the constraint of fixed microphone distributions across arms and rings. As the number of sensors on each ring can be adjusted as required, more microphones can be distributed internally in a spiral pattern to improve noise suppression, and additional microphones can be placed on the outer rings to improve the spatial directivity of the array. By balancing these two aspects, an optimal array configuration can be realised. Among the evaluated configurations, the annular array demonstrated the best balance between DR and BW. In this study, an annular array was chosen for sensor array implementation (Figure 9a).

## 4. Experiments and Results

### 4.1. Design of the Embedded System

The entire system was integrated onto two 12 cm square printed circuit boards (Figure 9): a sensor array and a control module. To minimize transformation errors when converting the spherical coordinate system of the microphone array to the Cartesian coordinate system of the camera during fusion and ensure localisation accuracy, the camera was mounted at the centre of the sensor array to guarantee the coincidence of their geometric centres.

The control module was primarily composed of two parts: a field-programmable gate array (FPGA), which was responsible for acquiring and processing sensor signals from the array, and an ARM processor, which handled beamforming calculations, captured visible-light images, performed imaging of gas leak points, and facilitated human–machine interaction. The communication method between the FPGA and ARM was via USB. The system hardware architecture is illustrated in Figure 10.

### 4.2. Calibration Method for Gas Leakage Rate

To simulate the gas leak source, a gas leak simulation setup was constructed, where an air compressor supplied the gas and a pressure-reducing valve regulated the pressure for precise control. Leak holes with apertures of 0.1 mm to 0.5 mm were fabricated to induce gas leakage. Because of the small aperture of the holes, which made precise calibration difficult, the bubble method was employed to calibrate the gas leakage rate, as shown in Figure 11. Because of the pressure difference across the leak hole, bubbles were continuously produced in the water-filled beaker. At the end of the timed interval *t*, the volume *V* of the bubbles collected in the graduated cylinder was read. The gas leakage rate *Q* of the leakage hole was calculated as follows:(11)$$Q=\frac{V}{t}$$

Here, the gas leakage rate was measured in millilitres per minute (mL/min).

### 4.3. Comparison of Algorithm Performance

Several approaches—including beamforming, near-field acoustic holography (NAH) [25], and the recently developed sparse reconstruction methods—can all achieve sound source localisation. To enable efficient long-range detection, the acoustic signal model was established under the far-field assumption, thereby rendering NAH, which relies on a near-field model, unsuitable for this application scenario. Given the high likelihood of secondary leaks during real-world detection, some sparse reconstruction methods based on the single-source assumption (e.g., LASSO-CS and OMP-CS) fail to meet the requirements for multisource discrimination. Therefore, this study selected two representative sparse reconstruction methods—Simultaneous Orthogonal Matching Pursuit (SOMP) [26] and Sparse Bayesian Learning (SBL) [27]—applying them to gas leak localisation and comparing their performance with conventional beamforming (CBF).

All algorithms were simulated in the MATLAB 2022b environment. The test data were collected using an annular array designed in this study, capturing acoustic signals from three gas leaks. The three algorithms were evaluated in terms of time complexity, space complexity, and localisation accuracy, as listed in Table 2.

M denotes the number of microphone channels, N represents the total number of spatial scanning points, L is the number of snapshots, T refers to the maximum number of iterations in the SOMP algorithm (set to 10 in this study), and K denotes the number of iterations in the SBL algorithm (set to 100 in this study).

In the multiple gas leak localisation task, all three algorithms demonstrated localisation errors within 5 pixels, indicating that they met the accuracy requirements for practical applications. Given that the required localisation accuracy can be achieved, algorithm selection should prioritize low time and space complexity to facilitate deployment on embedded system. As shown in Table 2, when the scanning resolution was 45 × 60, the runtimes of SOMP and SBL were approximately 25 times and 10,000 times longer than that of CBF. At a higher resolution of 240 × 320, SBL failed to complete execution due to excessive resource consumption, while SOMP’s runtime increased to roughly 30 times that of CBF—indicating a significant computational burden at high resolutions. These results clearly demonstrate that both SOMP and SBL are unsuitable for deployment on resource-constrained embedded systems.

Considering both localisation accuracy and computational efficiency, this study ultimately adopted CBF as the localisation algorithm for subsequent system implementation, ensuring a balance among accuracy, feasibility, and real-time performance on an embedded system.

### 4.4. Single-Point Gas Leak Test

The sensitivity of the proposed method and developed system in detecting minor gas leaks was verified by conducting experiments to determine the minimum detectable gas leak rate at various testing distances. At each test distance, the supply pressure from the pressure-reducing valve was gradually increased from a low initial value until the gas leaking from the leak holes was detected and localised. The current gas leak rate was then calibrated using the bubble method. During the testing, the gas leak holes were oriented to face the microphone array to ensure direct emission.

For a leak hole with an aperture of 0.1 mm, the minimum detectable and locatable gas leakage rate was 24 mL/min at close range and increased to 45 mL/min at a distance of 4 m (Table 3 and Figure 12). This indicates that the developed system can detect minor gas leaks, satisfying the sensitivity requirements necessary for maintenance personnel to replace or repair leaking containers. The methodology proposed in this study, along with the designed system, meets the sensitivity criteria for practical applications.

The accuracy of the proposed method and developed system in localising minor leaks was verified by conducting experiments to localise minor gas leaks at different positions at various test distances. The localisation coordinates were obtained and compared with the visually identified leak coordinates to calculate the localisation error. The system constructed in this study provided visual localisation; hence, a two-dimensional image coordinate system was established with the centre of the visualisation image as the origin and a display resolution of 800 × 600 pixels as a reference. Five gas leak source positions were specified: (0, 0), (−200, +150), (+200, +150), (+200, −150), and (−200, −150). The localisation coordinates estimated by the algorithm for different test distances are shown in Figure 13.

Using the test distance *d* and the field of view (*FOV*) of the camera, a geometric relationship was established between the image and camera coordinate systems.(12)$$\tilde{e}=\bar{e}\frac{d\cdot tan(FOV/2)}{500}$$

The localisation error $\bar{e}$ in the image coordinate system is calculated using Equation (12) to obtain the actual localisation error $\tilde{e}$; the results are listed in Table 4.

The localisation error in the image coordinate system did not change significantly for the gas leak sources at different test distances (Table 4), indicating that there exist unavoidable systematic errors (e.g., geometric calibration errors). As the test distance increased, the field of view expanded, and the actual localisation error increased proportionally with the increasing field of view (Table 4 and Figure 14). A localisation error of 0.68 cm was achieved at test distances of 1 m, 1.40 cm at 2 m, and 2.34 cm at 3 m. The experimental results showed that the developed system can accurately locate minor gas leaks and satisfy the precision requirements for maintenance personnel to replace or repair leaking containers. The proposed method and developed system can meet the accuracy requirements for practical minor leak detection.

### 4.5. Multi-Point Gas Leak Test

Given the high likelihood of encountering secondary leak points in real-world detection scenarios, it is essential to detect multiple gas leak points. To test the capability of the proposed method in addressing the challenges of detecting multiple gas leak sources, a simulation of a multi-point gas leak scenario was conducted. A gas leak pipeline system with small holes was constructed, in which the apertures of the leak holes ranged from 0.1 to 0.5 mm (Figure 15). The internal air pressure in the pipeline was adjusted using a pressure-reducing valve, and the pipeline was secured to a stand.

The experimental results highlight the ability of the system to detect and localise multiple gas leaks from diverse perspectives, effectively demonstrating stable performance in multiple gas leak detection and localisation (Figure 16).

To verify the localisation performance (resolution) of the proposed method and developed system in scenarios involving multiple gas leaks, a simulation setup capable of altering the spacing between leak holes was designed and implemented for a dual gas leak simulation (Figure 17). The dual leak holes were supplied with gas independently, enabling adjustment of the pressure-reducing valves for each leak hole to an appropriate pressure for detection.

With the dynamic range of the spatial power spectrum fixed at 2 dB, the lateral distance between the dual gas leak sources was gradually increased incrementally at various test distances. The approximate tangency of the boundaries of the dual beams in the imaging served as the resolution criterion for the spatial spectrum to resolve dual sources. The lateral distance between the centres of the dual leak holes at this point was recorded as the minimum resolvable spacing using this algorithm. The detection and localisation images of the dual gas leak sources are shown in Figure 18, and the results are presented in Table 5.

The leakage rate of the left leak hole was 303 mL/min, whereas that of the right leak hole was 209 mL/min (Figure 18). The algorithm can effectively distinguish among multiple gas leak sources (Table 5). At a 0.5 m test distance, it can distinguish dual leak sources spaced 5 cm apart, 1 m and 8 cm apart, 2 m and 18 cm apart, and 3 m and 22 cm apart. As the test distance increases and the field of view expands, each pixel corresponds to a larger distance, thereby partially reducing the resolution capability of the algorithm. The experimental results demonstrate that the developed system can locate multiple gas leak sources with different leakage rates and has good resolution capability, meeting the resolution requirements for maintenance personnel to replace or repair leaking containers. The proposed method and the developed system can satisfy the resolution requirements for practical multiple gas leak detection and localisation.

### 4.6. Real-Time Performance of Embedded System

This study also made significant efforts to enhance the real-time performance of the embedded system. During the code development process, several optimisation measures were implemented. The CPU cache hit rate was enhanced, the OpenMP parallel programming framework was used to optimise the utilisation of the resources of multicore CPUs, optimised compilation was employed to facilitate the vectorisation of matrix operations, and the clock speed of the ARM chips was increased. Within the scope of the algorithmic implementation, increasing the step size for beamforming angular scanning calculations diminished the computational burden. This reduction leads to reduced processing time and improved refresh rates. However, this comes with a trade-off in image resolution. To address this trade-off, this study employed image interpolation techniques to enhance image resolution. This enhancement renders further increasing the step size in angular scanning calculations possible, effectively balancing the computation time and image quality requirements. To further enhance the refresh rate, the precision of the computational process was reduced to an acceptable float type.

Finally, excluding the impact of other tasks, the computation time for single beamforming was 87 ms. This performance enables the practical application of the proposed system in time-sensitive industrial environments.

## 5. Conclusions

This study proposes a minor and multiple gas leak localisation imaging method to provide a universal, non-contact, efficient detection approach for potential gas leaks in pressure vessels, pipelines, valves, connectors, and similar components. Beamforming algorithms were employed to address the challenges associated with detecting multiple gas leaks, and cascaded filters were used for signal pre-processing in minor gas leak detection. The proposed method maintains phase alignment precision while significantly reducing resource utilization, enabling real-time processing on embedded platforms. Moreover, a series of image-processing techniques were utilised to achieve the visualisation of gas leak sources. Various geometric configurations were simulated and compared based on the performance metrics, with the annular array featuring the strongest noise suppression performance selected to enhance the system’s capacity in minor gas leak detection.

To verify the detection and localisation performance of the proposed method and developed system, a series of experiments were conducted. The results demonstrated that the system can detect minor gas leaks down to 24 mL/min and achieve high-precision localisation with an error of 0.68 cm at a test distance of 1 m. It can also reliably detect and localise multiple gas leaks from various angles and distinguish between dual gas leak sources spaced 8 cm apart at a test distance of 1 m. Moreover, the system exhibits real-time performance capabilities suitable for field deployment. This study demonstrates the feasibility of ultrasonic imaging for leak detection, offering a foundation for scalable, embedded sensing solutions with potential applications in HVAC systems, fuel lines, and industrial robotics.

However, this system has some limitations. Future work will focus on optimising the localisation algorithms through the integration of adaptive methods tailored for array signals to enhance robustness against environmental noise. Additionally, plans are in place to increase the number of microphones in the array to enhance minor gas leak detection capability.

## Figures and Tables

**Figure 1 sensors-25-03190-f001:**
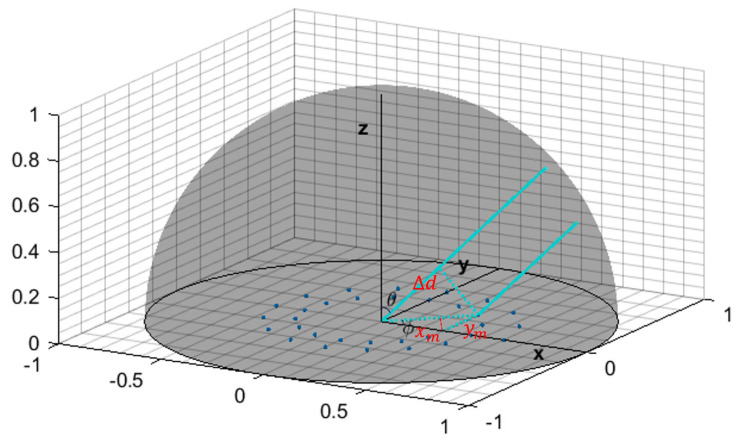
Definition of the scanning angle $\left(\theta ,\phi \right)$ [18].

**Figure 2 sensors-25-03190-f002:**
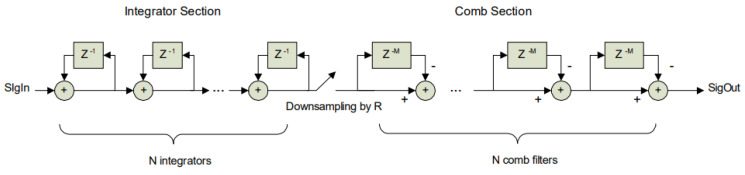
Structure of the CIC filter.

**Figure 3 sensors-25-03190-f003:**
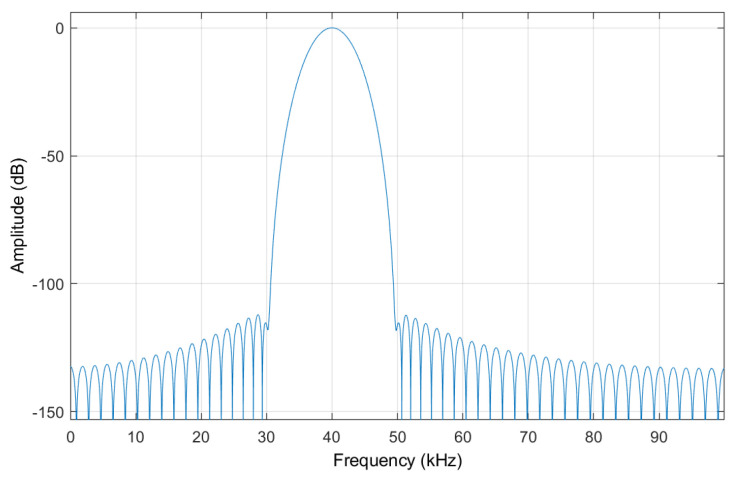
Amplitude–frequency response curve of the Chebyshev bandpass filter.

**Figure 4 sensors-25-03190-f004:**
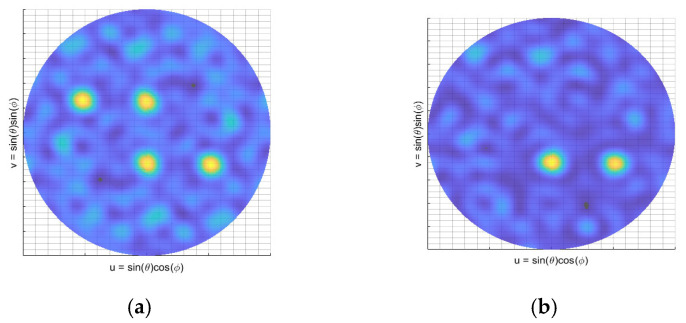
Role of the Hilbert transform: (**a**) central symmetry in the spatial power spectrum; (**b**) elimination of central symmetry post Hilbert transform.

**Figure 5 sensors-25-03190-f005:**
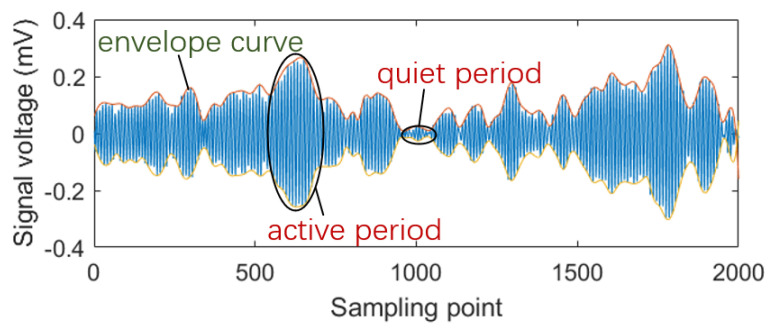
Turbulence alternating between an “active period” and a “quiet period”.

**Figure 6 sensors-25-03190-f006:**
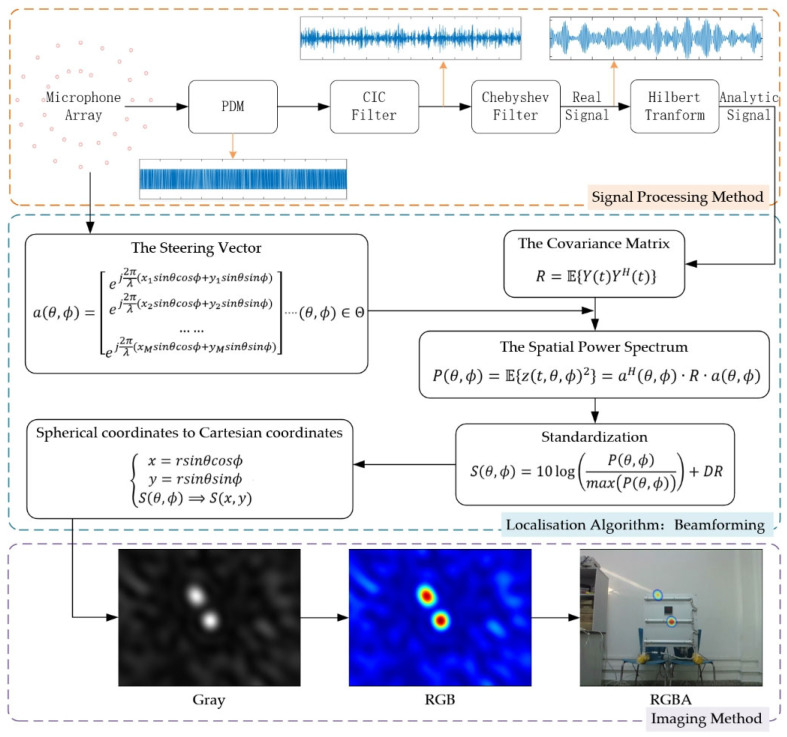
Overall architecture diagram of the signal processing pipeline from acquisition to visualization.

**Figure 7 sensors-25-03190-f007:**
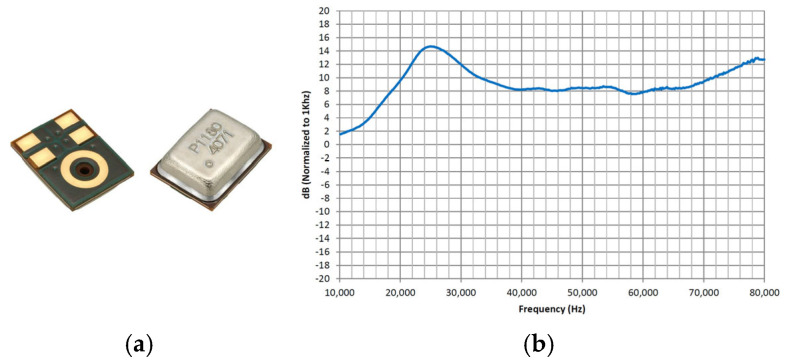
SPH0641LU4H-1: (**a**) photograph of SPH0641LU4H-1; (**b**) typical ultrasonic response normalised to 1 kHz.

**Figure 8 sensors-25-03190-f008:**
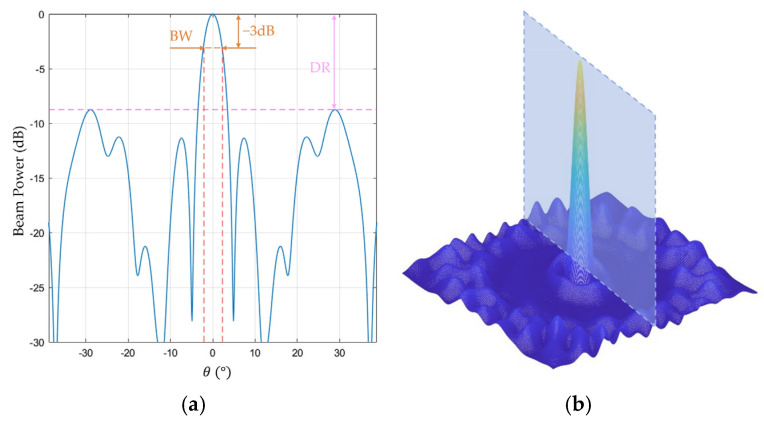
Array performance evaluation: (**a**) definition of BW and DR; (**b**) cross-section of the beam spatial power spectrum.

**Figure 9 sensors-25-03190-f009:**
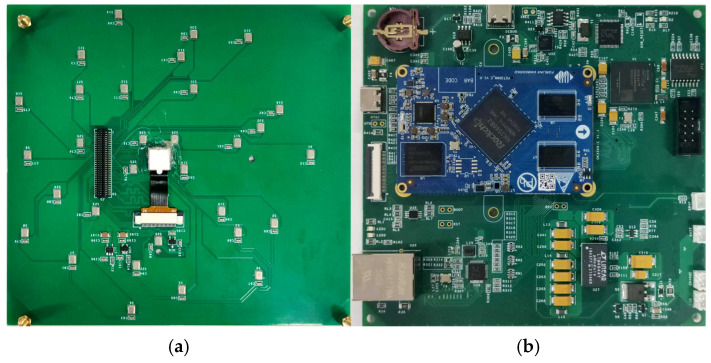
Photographs of the hardware system: (**a**) sensor array; (**b**) control module.

**Figure 10 sensors-25-03190-f010:**
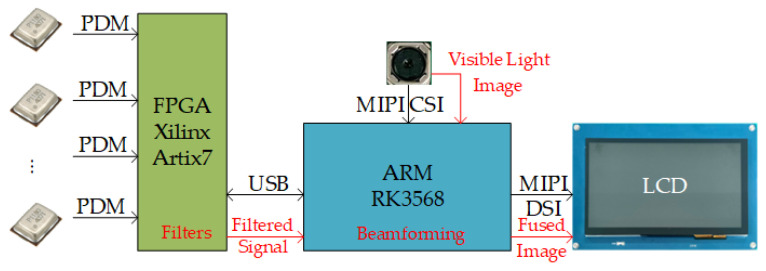
System hardware architecture.

**Figure 11 sensors-25-03190-f011:**
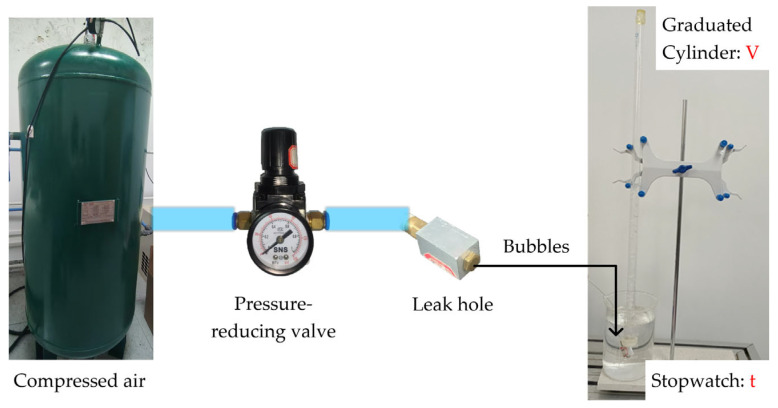
Schematic of the bubble method used to calibrate gas leakage rate.

**Figure 12 sensors-25-03190-f012:**
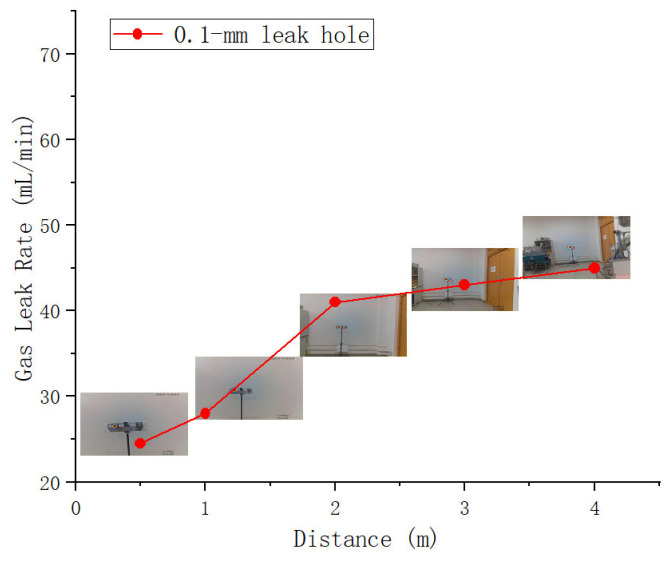
Minimum detectable gas leakage rates and corresponding test distances.

**Figure 13 sensors-25-03190-f013:**
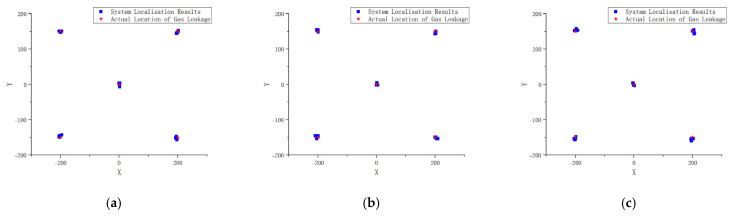
Localisation results of gas leak detection at different test distances: (**a**) 1 m; (**b**) 2 m; (**c**) 3 m.

**Figure 14 sensors-25-03190-f014:**
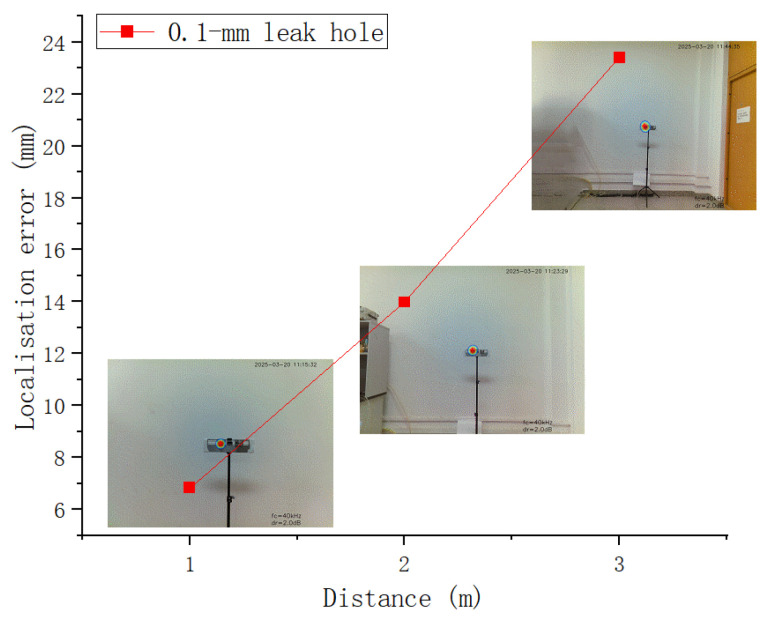
Localisation errors of gas leak detection at different test distances.

**Figure 15 sensors-25-03190-f015:**
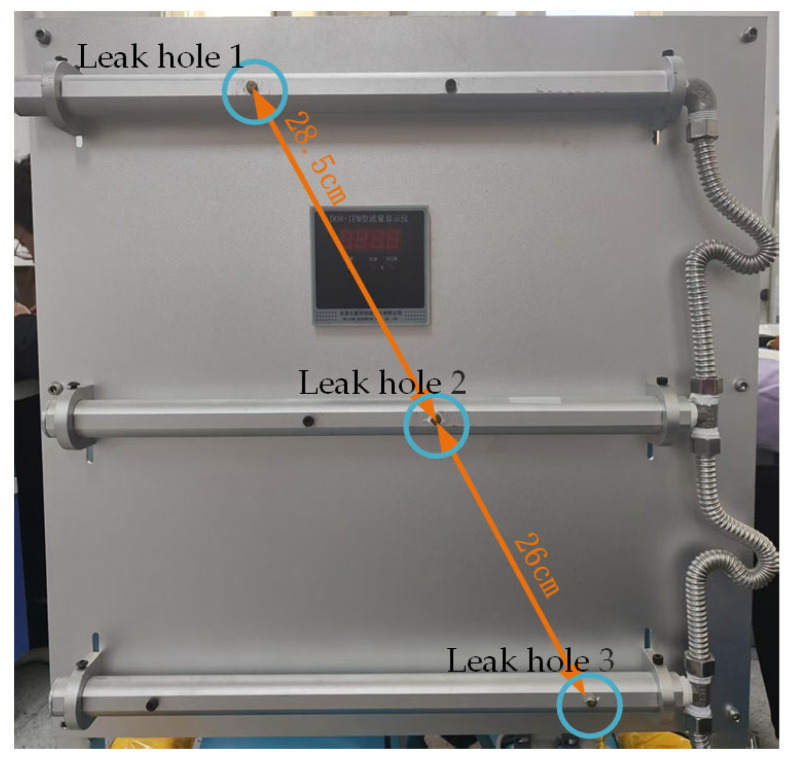
Gas leak pipeline system with multiple small holes.

**Figure 16 sensors-25-03190-f016:**
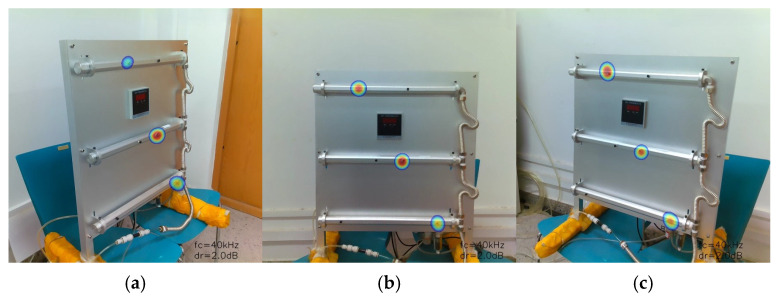
Detection and localisation of multi-point gas leaks from diverse perspectives. (**a**) Left-side perspective; (**b**) Front perspective; (**c**) Right-side perspective.

**Figure 17 sensors-25-03190-f017:**
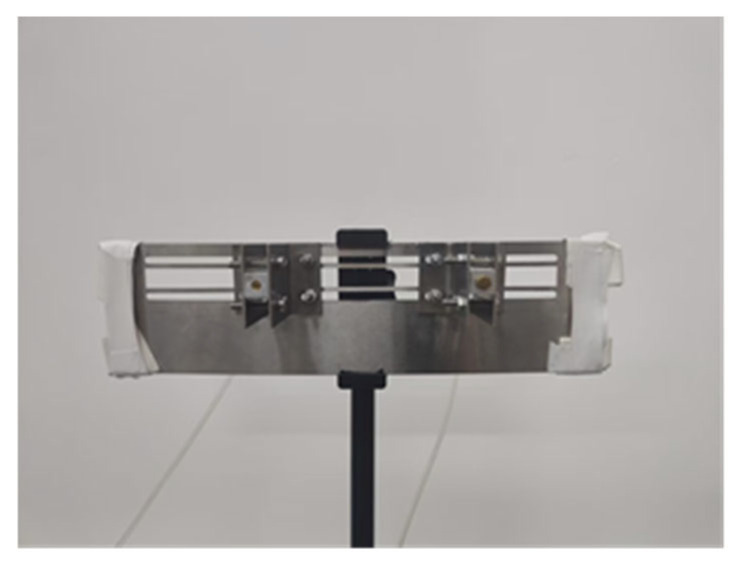
Adjustable dual gas leak simulation setup.

**Figure 18 sensors-25-03190-f018:**
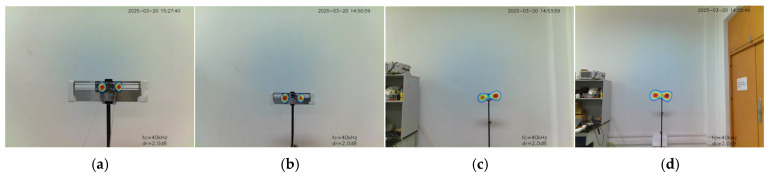
Dual localisation images of dual gas leaks at different test distances: (**a**) 0.5 m; (**b**) 1 m; (**c**) 2 m; (**d**) 3 m.

**Table 1 sensors-25-03190-t001:** Simulation results.

Array Types	BW	DR (dB)	DR Variance
Ring	3.90°	8.97	1.9784
Archimedean spiral	4.50°	8.06	0.0089
Dougherty log-spiral	4.10°	8.21	0.0275
Arcondoulis spiral	3.90°	8.01	0.0973
Rose spiral	4.70°	8.81	0.0884
Sunflower	3.50°	5.54	0.0001
Underbrink array	4.50°	7.39	0.0001
Annular array	4.50°	9.32	0.0012

**Table 2 sensors-25-03190-t002:** Comparison of CBF, SOMP, and SBL algorithms.

Resolution	Algorithm	Time Complexity	Space Complexity	Runtime	Localisation Error (Pixels)
45 × 60	CBF	O(M·N)	O(N)	0.003050 s	0.67
	SOMP	O(T·M·N)	O(N·L)	0.076395 s	0.67
	SBL	O(K·M·N^2^)	O(N^2^)	32.915978 s	0.34
240 × 320	CBF	O(M·N)	O(N)	0.046363 s	4.485
	SOMP	O(T·M·N)	O(N·L)	1.521570 s	4.955
	SBL	O(K·M·N^2^)	O(N^2^)	Out of Memory	Out of Memory

**Table 3 sensors-25-03190-t003:** Minimum detectable gas leakage rates across test distances.

Distance	0.5 m	1.0 m	2.0 m	3.0 m	4.0 m
0.1 mm Leak hole	24 cm^3^/min	28 mL/min	41 mL/min	43 mL/min	45 mL/min

**Table 4 sensors-25-03190-t004:** Localisation errors across test distances.

Distance	1.0 m	2.0 m	3.0 m
Image localisation error	4.70	4.81	5.37
Actual localisation error	0.68 cm	1.40 cm	2.34 cm

**Table 5 sensors-25-03190-t005:** Minimum resolvable spacing at different test distances.

Distance	0.5 m	1.0 m	2.0 m	3.0 m
Minimum Spacing	5 cm	8 cm	18 cm	22 cm

## Data Availability

Data are contained within the article.

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
