# Peer review of "Ultrasonic Beamforming-Based Visual Localisation of Minor and Multiple Gas Leaks Using a Microelectromechanical System (MEMS) Microphone Array"

_sensors, 2025, doi:10.3390/s25103190_

Round 1

Reviewer 1 Report

Comments and Suggestions for Authors

This paper presents an integrated ultrasonic beamforming system for visual gas leak localization. Several critical issues should be addressed to enhance its technical credibility:

  1. The manuscript lacks comparative experiments with existing methods mentioned in the literature. Without such baselines, it is hard to justify the claimed improvements in accuracy.
  2. It lacks an overall architecture diagram illustrating the full pipeline from signal acquisition, pre-processing, beamforming, image fusion, to final visualization.
  3. In real industrial environments, factors such as multipath interference and structural vibration noise have significant impact on performance. These aspects are not analyzed in the manuscript, raising concerns about the practical deployment of the system.
  4. The system claims sub-centimeter localization accuracy (e.g., 3.42 mm at 1 meter), yet it does not include any explicit geometric calibration between the camera and the microphone array. 
  5. Several figures lack clarity, e.g., no legend or labeling in Figure. 13.

Author Response

Dear Editors and Reviewers:

Thank you for your letter and for the reviewers’ comments concerning our manuscript ID sensors-3593581, entitled “Ultrasonic Beamforming-Based Visual Localisation of Minor and Multiple Gas Leaks Using a MEMS Microphone Array”. Those comments are all valuable and very helpful for revising and improving our paper, as well as the important guiding significance to our researches. We have studied comments carefully and have made correction which we hope meet with approval. Revised portion are marked in a yellow highlight tool in the paper. The main corrections in the paper and the responds to the reviewer’s comments are as flowing:

Response to the reviewer’s comments:

Comments 1: The manuscript lacks comparative experiments with existing methods mentioned in the literature. Without such baselines, it is hard to justify the claimed improvements in accuracy.

Response 1: Thanks for your valuable comment. We have added comparative experiments with existing algorithms in Section 4.3 (lines 327 to 362). The results demonstrate that the CBF approach achieves satisfactory localization accuracy while requiring significantly fewer spatial resources and shorter computation time than SOMP and SBL. Detailed quantitative comparisons are presented in Table 2.

It is important to clarify that our study does not aim to demonstrate superior accuracy compared to state-of-the-art algorithms. Instead, we propose a method that balances localisation precision with hardware feasibility, ensuring acceptable accuracy for practical applications while reducing computational complexity to embedded hardware platforms. This design prioritizes portability and real-time capability, which are critical for developing compact gas leak detection systems in industrial environments.

Comments 2: It lacks an overall architecture diagram illustrating the full pipeline from signal acquisition, pre-processing, beamforming, image fusion, to final visualization.

Response 2: Thanks for the valuable comment. We have added it to Figure 6 and included a portion of description (lines 241 to 250). We hope this will be helpful for you to understand the overall architecture diagram.

Comments 3: In real industrial environments, factors such as multipath interference and structural vibration noise have significant impact on performance. These aspects are not analyzed in the manuscript, raising concerns about the practical deployment of the system.

Response 3: Thanks for your valuable comment. We are sorry that although we tried our best, we still cannot clearly specify the extent of the impact of multipath interference or structural vibration noise on the system. However, the acoustic energy generated by industrial machinery such as motors, compressors, and exhaust fans concentrated below 20 kHz, while the noise from electronic circuits tends to occupy higher frequencies. To suppress these noises, we have employed a 100th-order Chebyshev band-pass filter. The amplitude-frequency response curve of this filter is added in Figure 3, demonstrating excellent passband isolation and out-of-band attenuation effects. This addition has been revised in Section 2.2, “Signal Processing Method” (lines 177–186).

Comments 4: The system claims sub-centimeter localization accuracy (e.g., 3.42 mm at 1 meter), yet it does not include any explicit geometric calibration between the camera and the microphone array.

Response 4: We are very sorry for our negligence of the interpretation. To minimize transformation errors when converting the spherical coordinate system of the microphone array to the Cartesian coordinate system of the camera during fusion and ensure localisation accuracy, the camera is mounted at the centre of sensor array to guarantee coincidence of their geometric centres. The statement has been added in lines 301 to 304.

In addition, we have corrected the error in the localisation error formula (12). Specifically, the actual distance of the field of view, which is d⋅tan(FOV/2), corresponds to 500 pixels (with a field of view resolution of 800×600). Therefore, the calculated localization errors are 6.83 mm at test distance of 1 m, 13.98 mm at 2 m, and 23.40 mm at 3 m. In the paper, we converted these errors to the centimeter unit, which provides more practical reference significance for real-world applications.

Comments 5: Several figures lack clarity, e.g., no legend or labeling in Figure. 13.

Response 5: We are very sorry for our negligence of the clarity. The legends have been added to Figure 13.

We tried our best to improve the manuscript and made some changes in the manuscript. These changes will not influence the content and framework of the paper.

We appreciate for your warm work earnestly, and hope that the correction will meet with approval.

Once again, thank you very much for your comments and suggestions.

Sincerely,

Tao Wang

School of Automation, Beijing Institute of Technology

Reviewer 2 Report

Comments and Suggestions for Authors

A very clear article with very interesting results but where there are two fundamental points that need to be properly explained. The first is that it performs exactly the Hilbert transform in the processing scheme by inserting figures on the spectrum of the real signals and the spectrum of the Hilbert transform. What is the advantage it provides in relation to performing a Fourier transform? The Hilbert transform of a signal in the time domain is a signal in the time domain where the phase of the positive frequencies is 90º out of phase. Does the auto-noise aim to obtain the analytical signal? This issue is described in a very superficial way  ( lines 178 to 188 )
Secondly, the paper does not describe what the beamforming algorithm to be used will be, ‘Upon completing the beamforming calculations’ line 219 ... where is the beamforming carried out, in what domain? temporal or frequency? What algorithm is it: deterministic, adaptive, statistically optimal? How do the receivers calculate the phase shift after the Hilbert transform? Although I have researched these two topics thoroughly, I cannot assess whether this publication is innovative or is a simple implementation of an ultrasonic band array, where there are many previous publications

Author Response

Dear Editors and Reviewers:

Thank you for your letter and for the reviewers’ comments concerning our manuscript ID sensors-3593581, entitled “Ultrasonic Beamforming-Based Visual Localisation of Minor and Multiple Gas Leaks Using a MEMS Microphone Array”. Those comments are all valuable and very helpful for revising and improving our paper, as well as the important guiding significance to our researches. We have studied comments carefully and have made correction which we hope meet with approval. Revised portion are marked in a yellow highlight tool in the paper. The main corrections in the paper and the responds to the reviewer’s comments are as flowing:

Response to the reviewer’s comments:

Comments 1: The first is that it performs exactly the Hilbert transform in the processing scheme by inserting figures on the spectrum of the real signals and the spectrum of the Hilbert transform. What is the advantage it provides in relation to performing a Fourier transform? The Hilbert transform of a signal in the time domain is a signal in the time domain where the phase of the positive frequencies is 90º out of phase. Does the auto-noise aim to obtain the analytical signal? This issue is described in a very superficial way  ( lines 178 to 188 )

Response 1: Thanks for your valuable comments. We have redescribed this section (lines 187–205) to enhance clarity, and we hope this improves your understanding.

To achieve precise phase delay, this study defines the steering vector in the frequency domain. Since sensor signals contain only real components, their frequency spectra exhibit symmetric positive and negative frequency components, leading to central symmetry in direct beamforming (Figure 4a). Traditional methods employ a fast Fourier transform (FFT) to transform signals to the frequency domain for phase shifting, followed by weighted summation across frequency bands to obtain spatial power spectrum. This approach requires signal frame segmentation and imposes high resource consumption during multi-channel hardware synchronization.

In contrast, this study employs the Hilbert transform to introduce a 90° phase delay for real signals, directly constructing analytic signals that satisfy Equation (1) to suppress negative frequency components. This enables frequency-domain beamforming to be performed directly from time-domain signals, eliminating the central symmetry (Figure 4b). The Hilbert transform can be implemented using hardware-based FIR filters, supporting continuous input signals without the need for frame segmentation. This method minimizes resource usage, making it well-suited for real-time application deployments.

Comments 2: Secondly, the paper does not describe what the beamforming algorithm to be used will be, ‘Upon completing the beamforming calculations’ line 219 ... where is the beamforming carried out, in what domain? temporal or frequency? What algorithm is it: deterministic, adaptive, statistically optimal? How do the receivers calculate the phase shift after the Hilbert transform? Although I have researched these two topics thoroughly, I cannot assess whether this publication is innovative or is a simple implementation of an ultrasonic band array, where there are many previous publications

Response 2: Thanks for your valuable comments. We have added the overall architecture diagram—from signal acquisition, pre-processing, beamforming, image fusion, to final visualization—to Figure 6. We hope this will be helpful for you to understand the overall architecture diagram.

Additionally, we present Equation (8) to illustrate the phase-shifting principle of the Hilbert transform. By constructing analytic signals via the Hilbert transform, we directly compute the covariance matrix to implement frequency-domain beamforming.

We propose a method that balances localisation precision with hardware feasibility, ensuring acceptable accuracy for practical applications while reducing computational complexity to embedded hardware platforms. This approach aims to provide a universal, non-contact, efficient visual localization method for gas leaks, applicable to potential leaks in pressure vessels, pipelines, valves, connectors, and similar components. A series of experiments demonstrate that the proposed method and developed system achieve favorable performance in terms of sensitivity, accuracy, stability, and resolution.

We tried our best to improve the manuscript and made some changes in the manuscript. These changes will not influence the content and framework of the paper.

We appreciate for your warm work earnestly, and hope that the correction will meet with approval.

Once again, thank you very much for your comments and suggestions.

Sincerely,

Tao Wang

School of Automation, Beijing Institute of Technology

Reviewer 3 Report

Comments and Suggestions for Authors

The author constructed an ultrasonic sensor array and built a gas leakage test simulation platform, which verified through leakage tests that the beamforming algorithm can achieve small leakage localization. In my opinion, the biggest problem with this paper is the lack of comparison with existing methods. Among the existing methods, compressed sensing, near-field acoustic holography, and sparse Bayesian learning can all achieve gas leak localization. The author should choose one or several algorithms to compare with the algorithm proposed in the paper. Without comparison, the innovation of the proposed algorithm cannot be proved.

Author Response

Dear Editors and Reviewers:

Thank you for your letter and for the reviewers’ comments concerning our manuscript ID sensors-3593581, entitled “Ultrasonic Beamforming-Based Visual Localisation of Minor and Multiple Gas Leaks Using a MEMS Microphone Array”. Those comments are all valuable and very helpful for revising and improving our paper, as well as the important guiding significance to our researches. We have studied comments carefully and have made correction which we hope meet with approval. Revised portion are marked in a yellow highlight tool in the paper. The main corrections in the paper and the responds to the reviewer’s comments are as flowing:

Response to the reviewer’s comments:

Comments 1: The author constructed an ultrasonic sensor array and built a gas leakage test simulation platform, which verified through leakage tests that the beamforming algorithm can achieve small leakage localization. In my opinion, the biggest problem with this paper is the lack of comparison with existing methods. Among the existing methods, compressed sensing, near-field acoustic holography, and sparse Bayesian learning can all achieve gas leak localization. The author should choose one or several algorithms to compare with the algorithm proposed in the paper. Without comparison, the innovation of the proposed algorithm cannot be proved.

Response 1: Thank you for these valuable comments. We have added Section 4.3 (lines 327–362), “Comparison of Algorithm Performance”, in which we analyze existing gas leak localisation methods and compare Sparse Orthogonal Matching Pursuit (SOMP) and Sparse Bayesian Learning (SBL) against the conventional beamforming (CBF) method. The results demonstrate that the CBF approach achieves satisfactory localization accuracy while outperforming SOMP and SBL in both spatial resource consumption and computation time. Detailed results are presented in Table 2.

It is important to clarify that our study does not aim to demonstrate superior accuracy compared to state-of-the-art algorithms. Instead, we propose a method that balances localisation precision with hardware feasibility, ensuring acceptable accuracy for practical applications while reducing computational complexity to embedded hardware platforms. This design prioritizes portability and real-time capability, which are critical for developing compact gas leak detection systems in industrial environments.

Table 2. Comparison of CBF, SOMP, and SBL Algorithms.

Resolution

Algorithm

Time Complexity

Space Complexity

Runtime

Localization Error (pixels)

45×60

CBF

O(M·N)

O(N)

0.003050 s

0.67

SOMP

O(T·M·N)

O(N·L)

0.076395 s

0.67

SBL

O(K·M·N²)

O(N²)

32.915978 s

0.34

240×320

CBF

O(M·N)

O(N)

0.046363 s

4.485

SOMP

O(T·M·N)

O(N·L)

1.521570 s

4.955

SBL

O(K·M·N²)

O(N²)

Out of Memory

Out of Memory

Where M denotes the number of microphone channels, N represents the total number of spatial scanning points, L is the number of snapshots, T refers to the maximum number of iterations in the SOMP algorithm, and K denotes the number of iterations in the SBL algorithm.

We tried our best to improve the manuscript and made some changes in the manuscript. These changes will not influence the content and framework of the paper.

We appreciate for your warm work earnestly, and hope that the correction will meet with approval.

Once again, thank you very much for your comments and suggestions.

Sincerely,

Tao Wang

School of Automation, Beijing Institute of Technology

Round 2

Reviewer 2 Report

Comments and Suggestions for Authors

The requested changes to the reviewed paper have been implemented exactly as indicated. Therefore, I fully agree with the revised version of the paper and consider it suitable for publication.

Reviewer 3 Report

Comments and Suggestions for Authors

The author has fully answered my question and I agree to publish this manuscript in its current form.